# Horizontal Gaze Palsy with Progressive Scoliosis with Overlapping Epilepsy and Learning Difficulties: A Case Report

**DOI:** 10.3390/brainsci12050613

**Published:** 2022-05-08

**Authors:** Emilia Matera, Maria Giuseppina Petruzzelli, Martina Tarantini, Alessandra Gabellone, Lucia Marzulli, Romina Ficarella, Paola Orsini, Lucia Margari

**Affiliations:** 1Department of Biomedical Sciences and Human Oncology, University Hospital “A. Moro”, Piazza Giulio Cesare 11, 70100 Bari, Italy; emilia.matera@uniba.it (E.M.); m.tarantini19@studenti.uniba.it (M.T.); alessandragabellonee@gmail.com (A.G.); lucia.marzulli@uniba.it (L.M.); lucia.margari@uniba.it (L.M.); 2Department of Basic Medical Sciences, Neuroscience and Sensory Organs, University Hospital “A. Moro”, Piazza Giulio Cesare 11, 70100 Bari, Italy; 3Medical Genetics Unit, Department of Human Reproductive Medicine, ASL Bari, Via Ospedale Di Venere 1, 70012 Bari, Italy; romina.ficarella@asl.bari.it (R.F.); paola.orsini@asl.bari.it (P.O.)

**Keywords:** HGPPS, learning difficulties, epilepsy

## Abstract

Horizontal gaze palsy with progressive scoliosis (HGPPS) is a rare congenital disease characterized by the absence of horizontal gaze movements, progressive scoliosis, and typical brain, cerebellum, and medullary malformations. Here we describe a pediatric HGPPS case with overlapping epilepsy and learning difficulties. A 6-year-old girl was admitted to the University Hospital of Bari for the onset of a tonic–clonic seizure. Electroencephalogram showed slow and sharp waves on the right side with the tendency to diffuse. Brain magnetic resonance imaging demonstrated malformations compatible with HGPPS. Ophthalmological and orthopedic evaluations confirmed conjugate horizontal gaze palsy and mild thoracolumbar scoliosis. Neuropsychological assessment attested normal intelligence but serious difficulties in reading and writing. In spite of neuroradiological malformations, visual difficulties, and spinal deformities, literature data are limited about any coexisting neurocognitive HGPPS symptoms. Literature data regarding such topics are very limited. If, on the one hand, the coexistence of such symptoms can be interpreted as occasional, it could support the idea that they could fall within a spectrum of HGPPS anomalies. In addition to the standard investigations, the activation of specific neuropsychological assessment programs could help interventions improve the specialist care and the quality of life of HGPPS patients.

## 1. Background

Horizontal gaze palsy with progressive scoliosis (HGPPS)—firstly reported in 1974 by Dretakis and Kondoyannis [1]—is a rare autosomal recessive congenital disease that is a part of the congenital cranial dysinnervation disorders, including a heterogeneous group of congenital nonprogressive innervation anomalies of the facial and ocular muscles. Both sexes are affected equally.

HGPPS is clinically characterized by the congenital absence of horizontal gaze movements and progressive scoliosis developing in childhood and adolescence [2].

The pathogenesis of the disease is linked to mutations in the ROBO3 gene located on chromosome 11q23–25, which is transmitted with autosomal recessive modality and encodes a transmembrane receptor Immunoglobulin expressed by Commissural (C) neurons in the developing embryonic spinal cord. C-neurons allow the coordination and integration of information from both sides of the body and are essential for multiple functions such as binocular vision, sound localization, or integrated sensorimotor responses. ROBO3 is involved in axon guidance, regulation of midline crossing of posterior brain axons, the direction of cell migration, location of lateral longitudinal pathways, assembly of the cytoskeleton, and regulation of growing axons [3].

Although more than 43 mutations impeding the decussation process during weeks 15 to 19 of the development of the central nervous system have been identified in different coded protein domains [4], there are cases in which no mutations of ROBO3 have been found. For this reason, it has been hypothesized that the phenotypic manifestations of HGPPS may be caused by abnormalities in ROBO3 splice variant expression, mutations of a gene different from ROBO3, or some environmental or epigenetic factors inhibiting the action of ROBO3 or its receptor product [5].

MRI images show distinctive brain, cerebellum, and medullary malformations such as brain stem and pontine hypoplasia, absence of facial colliculi, butterfly configuration of the medulla, and pontine cleft in the midline. Only in a few cases MRI images result was normal [6]. Several authors have used Diffused Tensor Imaging (DTI) to identify specific fiber tracts and their directionality in HGPPS and have shown the absence of major crossing pathways in the pons, midbrain, and corticospinal tract [7,8].

In HGPPS patients, defects in the medial longitudinal fasciculi and adjacent areas of the abducens nuclei and paramedian pontine reticular formation lead to horizontal gaze palsy [9]. Scoliosis is the most common reason for seeking medical care and causing disability, so much so that a surgical correction is necessary to reduce the angle of the scoliotic curve [10].

Most authors focus on the description of the pathogenesis, morphostructural alterations, and classic clinical features of HGPPS syndrome and occasionally report comorbid cognitive and neurological symptoms without further investigations. The objective of this study is, therefore, to describe an illustrative pediatric HGPPS case with overlapping epilepsy and learning difficulties and review the available literature data on this topic.

## 2. Case Report

A 6-year-old girl was admitted to the Child Neuropsychiatric Unit of the University Hospital of Bari for the onset of a tonic–clonic seizure soon after waking up, with loss of consciousness followed by falling and characterized by upward eye rolling, wheezing sounds, trunk and limb spasms, and urinary incontinence which gradually disappeared after three to five minutes. A previous seizure with the same features occurred about six months earlier.

She was the third child of consanguineous healthy parents, who were first cousins. A brief review of the child’s medical history did not reveal any specific problems with pregnancy, labor, and motor and linguistic development. She attended the first grade of primary school with difficulties in reading and writing.

Neurological clinical examination was made using Touwen’s Examination of the Child with Minor Neurological Dysfunction procedures [11]. The examination of the head showed swinging head movements during walking and running, as well as slight hyperextension during fixation of frontally placed objects that her parents recognized early in life. The evaluation of the extrinsic ocular muscles revealed absent eye conjugate horizontal movements with a preserved vertical gaze. The assessment of standing and walking showed difficulty maintaining an upright position and jumping on one limb on both right and left sides, clumsy gait, and synkinetic arm movements accentuated in amplitude. The vestibular tests (Romberg, tandem, and star-shaped gait) and cerebellar tests (index–nose test with open and closed eyes) were not correctly performed. No other neurological deficits were present.

Blood tests (blood count, electrolytes, creatinine, urea, glucose, bilirubin, aspartate aminotransferase, alanine aminotransferase, gamma-glutamyl transpeptidase, albumin, cholesterol, triglycerides, lactate dehydrogenase, creatine phosphokinase, fibrinogen, thyrotropin, free thyroxin, free triiodothyronine, and C reactive protein) had values within the normal range. The interictal electroencephalogram (EEG) showed slow and sharp waves in the parietal, temporal, and occipital regions prevalent on the right side during sleep and wakefulness, with a sporadic tendency to diffuse (Figure 1). Treatment with Valproic Acid (15 mg/Kg/die) was therefore started.

Brain magnetic resonance imaging (MRI) demonstrated hypoplasia of the brainstem (Figure 2), a left thalamus volume larger than the right (Figure 3), an enlarged fourth ventricle, an anterior and posterior midline bulbar and pontine cleft (split-pons sign) (Figure 4), a butterfly configuration of the medulla (Figure 5), an occipital and parietal plagiocephaly on the right side, and a slight benign enlargement of the subarachnoid spaces in the frontotemporal area. There were no signal changes within the medulla or pons in any of the brain MRI sequences.

The ophthalmological and optometric evaluation revealed the absence of eye movements in conjugate horizontal movements but preserved in vertical gaze and convergence bilaterally, absence of bilateral abduction, and transient torsional nystagmus during adduction. There were also orthophoria at the Cover test, fusion at the Worth 4 dot test, uncertain stereopsis, normal ocular fundus, and visual acuity of 6/10 at the Monoyer chart.

Orthopedic examination and standard X-rays of the spine revealed a scoliotic attitude or “pseudo scoliosis” of the dorsal–lumbar spine associated with unlevel shoulders and left thoracic hump with Risser Grade 0. No lumbar salience or vertebral rotation signs were found (Figure 6). Unfortunately, we have no data on how scoliosis has changed over time to determine its progression because of the family’s lack of cooperation in carrying out the periodic clinical and instrumental investigations that this case required.

The genetic investigation confirmed the presence of a homozygous ROBO3 variant (c.3584_3588dup, p.Ile1197TrpfsTer44), which causes a shift in the reading window and a premature termination of the protein. Such ROBO3 variant is not found in the gnomAD exomes database, in ClinVar database, and in PubMed.

Neuropsychological assessment was performed through the administration of standardized tests. The Italian standardization of the Wechsler Preschool and Primary Scale of Intelligence-III [12] attested a Full-Scale Intelligence Quotient of 88, a Verbal IQ of 100, a Performance IQ of 89, and a Processing Speed Quotient of 70. The New MT reading tests for primary school classes 1–2 [13] revealed serious difficulties in reading comprehension, correctness, and speed of reading (score below the fifth percentile), in performing written tasks, though the patient recognized and wrote most letters/numbers and her name in capital letters. Oppositional and avoidant behaviors were evident when the child was asked to perform such school-type tasks. Child Behavior Checklist for ages 6–18 [14] showed no emotional and behavioral problems.

Such findings were suggestive of HGPPS syndrome associated with plausible focal epilepsy with focal to bilateral tonic–clonic seizures (ILAE 2017) [15] and learning difficulties which could suggest the hypothesis that HGPPS is a genetically determined syndrome associated with encephalopathy.

## 3. Discussion

We described the case of a female child patient with HGPPS and comorbid epilepsy and learning difficulties. To our knowledge, this is the first case in which, together with the standard clinical and instrumental assessment, a neuropsychological evaluation was also made in an HGPPS patient.

In our patient, we found conjugate horizontal gaze palsy associated with transient torsional nystagmus during adduction, mild thoracolumbar scoliosis, and neuroradiological signs including hypoplasia of the brainstem, split-pons sign, butterfly configuration of the medulla, enlarged fourth ventricle, and absence of the facial colliculi.

Since scoliosis and horizontal gaze palsy are the most obvious outward signs, which often justify the neuroradiological study, HGPPS patients may have variable oculomotor defects related to convergence, alignment, blinking, vestibulo-ocular responses, and tracking; in several cases, vertical gaze could be partially compromised too. Visual fields, pupil function, accommodation, anterior and posterior segments of the eye, and visual acuity are generally not severely impaired [16,17].

When better reported, progressive scoliosis is in the thoracic or thoracolumbar area [18] and presents with varying levels of severity that justify physical therapy, braces, and, more often, spinal surgery.

In cases where brain MRI is performed, the described lesions include butterfly-shaped malformation of the medulla, brainstem/pons/medulla oblongata/pyramids/vermis hypoplasia, reduction of cerebellar peduncles, absence of the facial colliculi protrusion, tent-shaped of the floor of the fourth ventricle split pons sign, and prominent inferior olivary nuclei; neuroimaging are normal in few patients [6,7,8,16,19,20].

Our patient, suffering from epilepsy, showed swinging head movements during walking and running, hyperextension of the head during front fixation, and gait, balance, and coordination abnormalities.

In spite of the previously described brain, cerebellum, and medullary malformations, visual difficulties, and spinal deformities, literature data are limited about any coexisting neurological and cognitive HGPPS symptoms, consequences, and limitations. No cases of comorbid epilepsy or electroencephalographic abnormalities are known. Some authors, however, reported the presence of other neurologic anomalies, such as trunk imbalance, altered walking, coordination and reflexes disturbances, intention tremor, headache, hearing impairment, and head movements as compensations for the horizontal gaze restriction (see Table 1) [6,16,19,21,22,23,24,25,26].

If, on the one hand, the coexistence of these symptoms could be interpreted as occasional, it could suggest a more widespread brainstem lesion in some HGPPS patients with the possibility that a spectrum of abnormalities might be included [7].

Although our patient had an overall normal cognitive function for age, her cognitive abilities showed considerable variation; lower scores were found in discrimination, visual memory, visual–motor coordination, attention, and concentration. The neuropsychological evaluation revealed an IQ within the average range, with significant differences between perceptual reasoning, verbal comprehension, and processing speed. The verbal abilities, as measured by the VIQ (100), were in the average range of intellectual functioning, suggesting adequate verbal reasoning/comprehension, acquired knowledge, and attention to verbal stimuli. The performance and processing speed abilities, as measured by PIQ (89) and PSQ (70), were in the low average range, indicating difficulties in fluid reasoning, spatial processing, perceptual organization, visual–motor integration, and especially in the ability to correctly scan, sequence, and discriminate simple visual information. Reading and writing were performed with considerable slowness, frustration, and avoidant behavior.

With respect to cognitive and learning abilities, most studies do not provide data; when some authors report normal intelligence or cognitive delay, they do not give information on the IQ value or the tests performed for its evaluation in patients with HGPPS. In the same way, it appears that specific tests investigating basic academic skills have never been administered (see Table 1) [6,7,8,16,23,24,27,28,29,30].

Reading and writing are visually mediated psychological processes. Reading requires precise coordination of both lower-level ocular motor processes (version, accommodation, and vergence) and higher-level nonocular motor processes (basic language skills, efficient processing speed, and intact executive functions such as working memory, attention shift, and strategic problems solving skills) through which the text is visually encoded, represented in abstract orthographic form, lexically elaborated, and understood in syntax and meaning. The act of writing involves multifaceted cognitive processes, including linguistically related ones, assignment of the meaning of visual symbolic representations, eye–hand coordination, and high-level motor control [31,32]. It is, therefore, plausible that the oculomotion deficits and compensative head movements may affect the cognitive and academic performances of at least one part of HGPPS patients. We might also hypothesize that the morphostructural brain, cerebellum, and medullary alterations deriving from ROBO 3 gene mutation, essential for axons crossing the midline of the posterior brain and neuronal migration to the contralateral side during the development of the nervous system, could justify the neuropsychological deficit. Recent preliminary neuroanatomical and neuroimaging studies have, for example, led to the conceptualization that the functional role of the cerebellum may extend not only to the control of movement but also to various cognitive aspects, including language, affectivity, executive control, and probably other functional processes, through cerebro–cerebellar olive circuit systems [33,34]. Mutation of ROBO 3 may affect the way connections are organized in the brain and also determine cognitive and learning problems that, according to recent evidence, are not related to malformations of specific brain areas, as previously thought, but to the way the brain is wired. Understanding how this actually develops and thus causes difficulties is, however, still extremely complex [35]. These data do not exempt us from supporting the idea that neurocognitive alterations could also fall within the spectrum of HGPPS anomalies [7].

Although we did not focus on the psychopathology associated with HGPPS syndrome, as our patient exhibited only oppositional, avoidant behaviors likely secondary to school learning difficulties, there are very limited literature data on psychological and behavioral problems coexisting with such disease, generally referring to the emotional distress resulting from physical deformities secondary to scoliosis [27,28,29,30]. Typically, HGPPS prognosis may be assessed on a case-by-case basis because it depends on the severity of the signs, symptoms, and any associated complications. Scoliosis, in particular, could increase in severity and reduce mobility and quality of life if left untreated [36]. Other associated problems such as ocular motor deficits, school difficulties, and possible neurological comorbidities could have negative repercussions on self-esteem, social life, mood, the performance of daily activities, and therefore on the possible onset of defined psychopathological disorders.

## 4. Conclusions

A multidisciplinary approach is essential for early diagnosis, application of supportive therapies, and careful study of any associated clinical condition in HGPPS patients. In addition to brain MR, X-rays of the spine, and genetic study, the activation of specific neuropsychological investigation programs for each evolutionary phase could help interventions that have been neglected up to now, such as personalized teaching, psychological/psychiatric support paths, and any drug therapies improving the specialist care and the quality of life of these patients.

## Figures and Tables

**Figure 1 brainsci-12-00613-f001:**
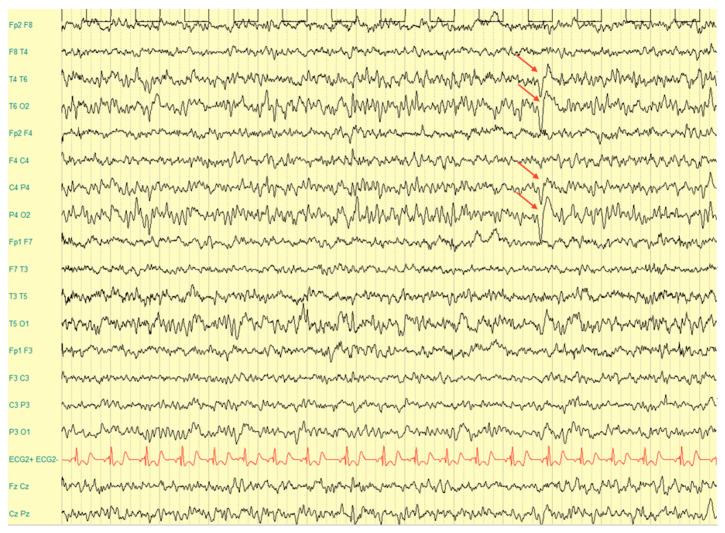
EEG recording showing slow and sharp waves in the temporal, parietal, and occipital regions prevalent on the right side.

**Figure 2 brainsci-12-00613-f002:**
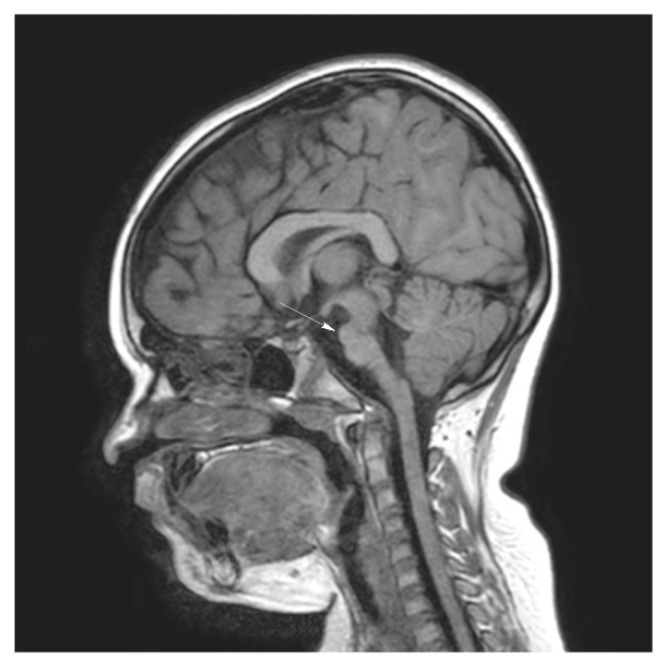
MRI brain imaging showing hypoplasia of the brainstem.

**Figure 3 brainsci-12-00613-f003:**
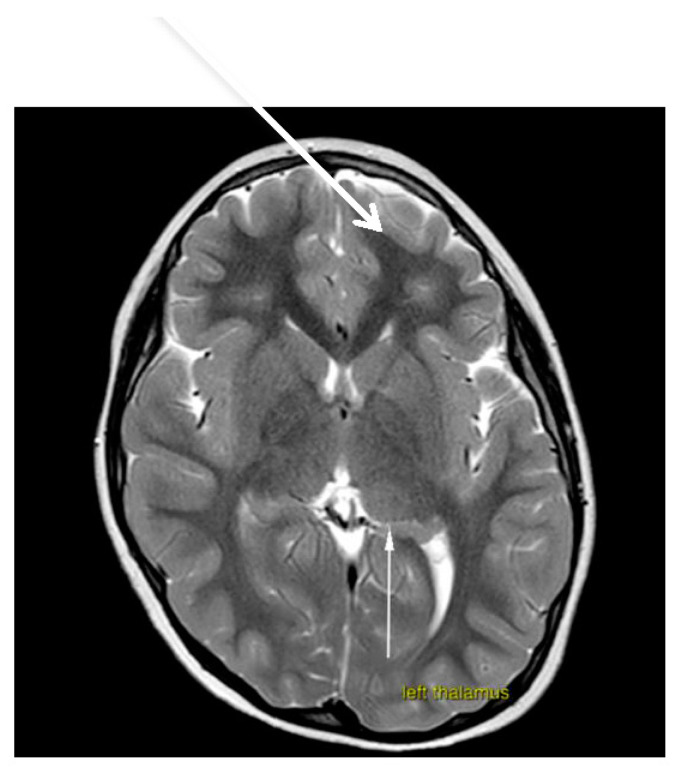
MRI brain imaging shows that the volume of the left thalamus is larger than the right.

**Figure 4 brainsci-12-00613-f004:**
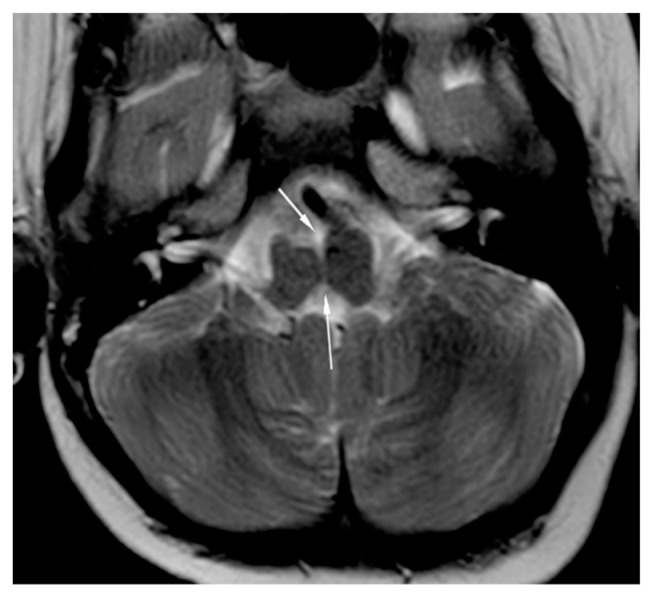
MRI brain axial T2-weighted image showing split pons sign.

**Figure 5 brainsci-12-00613-f005:**
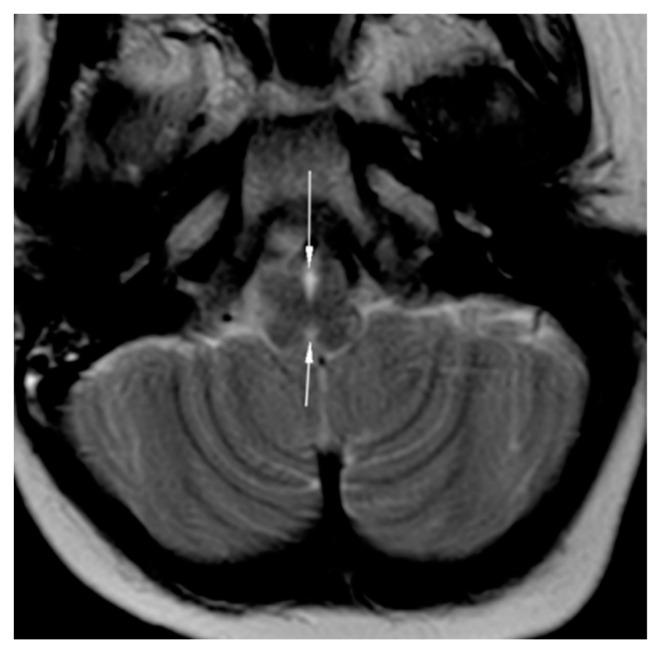
MRI brain axial T2-weighted image showing butterfly medulla oblongata.

**Figure 6 brainsci-12-00613-f006:**
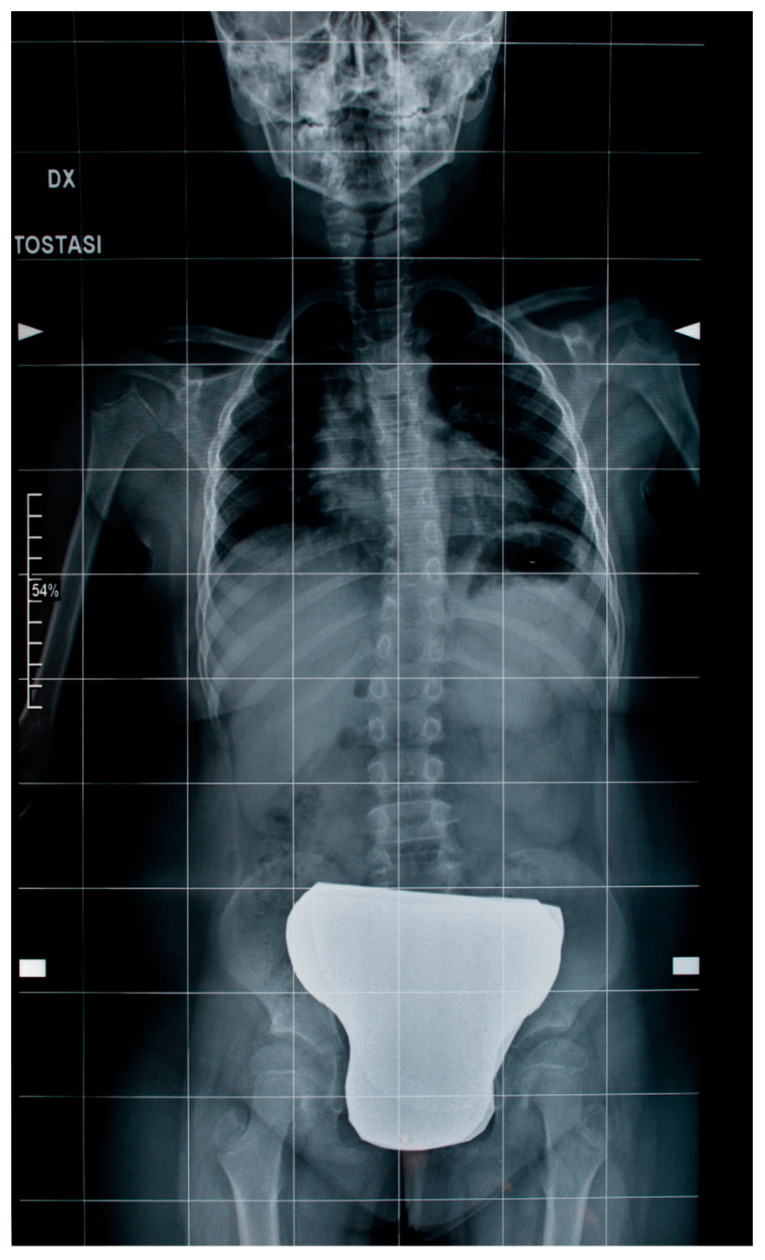
Standard X-rays of the spine revealed a scoliotic attitude of the dorsal–lumbar spine.

**Table 1 brainsci-12-00613-t001:** Literature review on the neurocognitive symptom in HGPPS syndrome.

Authors	Type of Study	P Number	HGP	Scoliosis	MRI Typical Findings	Cognitive Functioning	Neurological Symptoms
MacDonald DB et al., 2004 [27]	Case report	1	Y	Y	Y	Normal cognitive development	-
Lo B et al., 2004 [28]	Short report	3	Y	Y	Y	Cognitive delay (2 p)	-
Incecik F. et al., 2006 [6]	Brief communication	2	Y	Y (1 p)	Normal (1 p)	Normal cognitive development (all p)	Head tilt and titubation (1 p)
Amoiridis G et al., 2006 [21]	Case report	2	Y	Y	Y	-	Difficulty with tandem walking and performing alternating movements with the upper and the lower extremities simultaneously, Monolateral sensorineural hearing loss, Head jerks for looking left and right
Haller S et al., 2007 [22]	Case report	1	Y	Y	Y	-	Intention tremor, Unsteadiness on standing/jumping on one leg
Khan AO et al., 2008 [29]	Case report	1	Y	Y	Y	Normal cognitive development	-
Otaduy MCG et al., 2009 [7]	Case report	2	Y	Y (1 p)	Y	Cognitive delay (1 p)	-
Abu-Amero KK et al., 2011 [23]	Research report	4	Y	Y	Y (1 p)	Normal mental development (all)	Head turning in order to fixate with either eye
Volk AE et al., 2011 [20]	Research report	4	Y	Y	Y (1 p)	-	Head nodding (1 p)
Samoladas EP et al., 2013 [24]	Case report	1	Y	Y	Y	Normal cognitive development	Headache
da Silva de Magalhães MJ et al., 2014 [25]	Case report	1	Y	Y	-	-	Headache Bilateral hearing impairment
Handor H et al., 2014 [17]	Case report	4	Y	Y	Y (2 p)	-	Head tremor during gaze fixation attempts (3 p)
Fernandez-Vega Cueto A et al., 2016 [16]	Short communication	1	Y	Y	-	Normal mental development	-
Lin CW et al., 2018 [8]	Case report	1	Y	Y	Y	Normal mental development	-
Scortegagna FA et al., 2020 [30]	Case report	1	Y	Y	Y	Cognitive delay	-
Menon V et al., 2021 [26]	Case report	1	-	Y	Y	-	Trunk imbalance Titubation

p—patient, HGP—Horizontal Gaze Palsy, Y—yes.

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
