# Peer review of "Horizontal Gaze Palsy with Progressive Scoliosis with Overlapping Epilepsy and Learning Difficulties: A Case Report"

_brainsci, 2022, doi:10.3390/brainsci12050613_

Round 1
Reviewer 1 Report
The authors have considerably improved the manuscript and have included the suggestions made.
Author Response
Dear reviewer,
thank you for your comments.
Kind regards,
Emilia Matera

Reviewer 2 Report
The authors reported a case with horizontal gaze palsy with progressive scoliosis (HGPPS), plus epilepsy and learning difficulties as her additional pathological features. The pathogenesis and/or mechanism regarding epilepsy and learning difficulties are discussed in the manuscript, particularly aimed at elucidating their existence as the occasional comorbidity or pathological features associated with HGPPS. While the manuscript is quite informative, there are several points in need of clarification or revision.
(i) Diagnosis: Please provide brain MRI and spine X ray (at least) to support the description in the manuscript (line 120-123 and line 131-133). The important radiological features should be clearly marked and described in the Figure and the corresponding legend.
(ii) Description of imaging findings: Please revise brain and spine imaging description properly. For example, it is seldom mentioned as "medulla MRI" in both clinical and research field. A protruded disc of spine is not the result from a "brain" MRI. The asymmetry of iliac crests shall be clearly stated by measurement during physical examination or any imaging film. Moreover, the value (i.e., 3 "mm") is quite confusing and misleading. Such precise description is very important, particularly in a case whose genetic sequence is unidentified from the current HGPPS database.
(iii) Relevancy between learning difficulties with the current evaluation:
Please describe any morphological changes within cerebrum, in addition to radiological findings that are typical for HGPPS. Please elaborate the association between learning difficulty and EEG and/or brain MRI findings in depth (line 197-224), as brainstem-cerebellar circuits anomaly generally has limited impact onto epilepsy occurrence and cognitive deficits.
(iv) Description and interpretation for cognitive status: The authors stated "Although our patient had grossly normal cognitive function for age, her cognitive abilities showed considerable variation: lower scores were found in discrimination,....." (line 193-194). This sentence is quite confusing. Please modify it properly.
(v) Please mark clearly the finding of EEG (i.e., slow and sharp waves) and improve resolution of EEG data.
Reviewer 3 Report
The title corresponds to the content of the article.
Abstract: Contains the most important information about the case and the horizontal gaze palsy with progressive scoliosis (HGPPS) syndrome.
Keywords - well-chosen
Background: Provides detailed information on the complex etiopathogenesis of HGPPS and discusses the spectrum of symptoms. However, I do not like one sentence: "Scoliosis - that is the most common reason for seeking medical care and causing disability so much so that a surgical correction is necessary to reduce the angle of the scoliotic curve -seems not caused by an underlying disease of the muscle, spinal cord, or spine, but by neurological mechanisms involving the proprioception mediated by uncrossed corticospinal pathways and medial dorsal-lemniscal column, the postural balance, the pontine paramedian reticular formation, and the visual and vestibular reflex interaction [10]. I would avoid saying that in this case the scoliosis is not caused by a disease of the spinal cord, spine or muscles. I note that the pathway of pro-proceptive sensation (posterior column-medial lemniscus pathway) runs through the spinal cord, and the receptors of this pathway are found in the muscle. For example, we understand a muscle not only as a tissue, but as an organ made up of many tissues. This part of the sentence is unnecessary. Even if it is quoted after other authors, it requires a critical view.
Case report: Includes an insightful presentation of the case. However, there is no clear grouping of symptoms. Presence of indicator symptoms suggesting occurrence of the syndrome: coexistence of horizontal gaze palsy and scoliosis. While horizonatal gaze palsy is described in detail, scoliosis is not sufficiently characterized. At least the most basic data are missing: the Cobb angle for each curve of curvature and the bone age assessment based on the Risser test. Do you have any data on how scoliosis has changed over time - on this basis, you can determine the progression. If not, it should be taken into account that you do not have such information. Another group of the described symptoms is characteristic of damage to the central nervous system, which should be emphasized. At the end of the description, it can be emphasized that the HGPPS syndrome is a genetically determined syndrome associated with encephalopathy.
Discussion: the authors compare the symptoms of the presented syndrome to the characteristics of the syndromes presented in the literature. I do not consider it appropriate to describe the neurocognitive changes as ("minor", rather I see them as very important.
Conclusion: They are accurate and understandable.
Other: Prognosis for this group of patients is unknown: will they be independent in the future?
Round 2
Reviewer 2 Report
Please modify description of the figure legend properly as suggestions:
“Figure 1. EEG recording showing slow and sharp waves in the fronto-central regions prevalent on the right side”
- Two arrows are placed onto “T6-O2” and “P4-O2”; these two bipolar poles are not belonged to the fronto-central regions. The current figure legend is also not consistent with the manuscript description (line 116-117).
“Fig.2. MRI brain imaging showing hypoplasia of the brainstem”
- Please mark up the hypoplasia portion.
“Fig. 3. MRI brain imaging showing thalami asimmetry with left > right”
- It’s difficult to state which site is pathological based onto the current imaging data. It would be better to state the volume of the left thalamus is larger than the right, rather than stating asymmetry. If the authors are prone to state which site is pathological, it would be better to elaborate their interpretation by comparing an age/sex-matched control image.
“TW2 image” in Figure 4 and 5
- The term “T2-weighted images” or “T2WI” or “T2W images” would be more common in use. Considering the context of the current manuscript, it would be better to spell out this term.
Author Response
Please see the attachment

This manuscript is a resubmission of an earlier submission. The following is a list of the peer review reports and author responses from that submission.
Round 1
Reviewer 1 Report
Thank you for the opportunity to review this Case Report on HGPPS disease.
There are several formal aspects to improve:
- The font size must be the same throughout the document, check line 106.
- The table that appears must be numbered (Table 1). In addition, the appearance of the table could be improved so that it is more proportionate in the text.
On the other hand, in terms of methodology and content, highlight the following:
- Regarding line 82-83, what tool was used to measure the girl's development and what were the results?
- When you talk about "Neurological examination", what type of exam are you referring to? Could you explain it better?
- How did you measure gait, balance and coordination?
You have described a number of symptoms, but no specific assessment tools or results have been described.
They could also add the results of the electroencephalogram, to make the manuscript more visual, adding it as an image.
The optometric and ophthalmological evaluation tests, as well as the orthopedic examination to elucidate scoliosis, are also not described.
Reviewer 2 Report
Nicely presented manuscript, analyzing the case of a patient with horizontal; gaze palsy with progressive scoliosis, combined with overlapping epilepsy and learning difficulties.
This case represents a rare congenital disease but it is a well-described entity and this article is devoid of any novel information for the reader.
Round 2
Reviewer 1 Report
The authors have introduced the suggestions properly
Reviewer 2 Report
Dear Authors,
Thank you for your reply.
Actually, you have mentioned that 'focus on the neuropsychological aspects (school difficulties,
epilepsy), which are little investigated in the literature, and on the importance of considering the study
and treatment of such aspects which certainly have negative repercussions on overall functioning in
HGPPS patients, especially in developmental age.'
I strongly appreciate your comments. According to my point of view, the minimu prerequisite in order to effectively support this aspect of the study is to present a small case series with proper follow-up and not the clinical course of an isolated clinical case.